# GRF-LLM: Environment-Aware Wireless Channel Modeling via LLM-Guided 3D Gaussians

## Abstract

We present GRF-LLM, a transformative framework for environment-aware wireless channel modeling that synergizes 3D Gaussian splatting with large language models (LLMs). Traditional wireless channel modeling approaches face critical limitations: probabilistic models lack spatial granularity for complex multi-antenna systems, while deterministic methods like ray tracing require precise environmental priors that are often unavailable or inaccurate in practice. Our proposed method addresses these challenges by introducing LLM-guided initialization and optimization of 3D Gaussian primitives that dynamically encode both signal propagation characteristics and material-dependent attenuation effects. The framework leverages LLMs' semantic understanding capabilities to analyze environmental scenes and predict material properties, enabling intelligent placement and parameterization of virtual transmitters represented as 3D Gaussians. These primitives are processed through a differentiable rendering pipeline that reconstructs wireless radiation fields with unprecedented efficiency and accuracy. The integration of LLM guidance enables automatic adaptation to diverse environmental conditions without requiring extensive manual parameter tuning or detailed material property databases. Through comprehensive experimental validation on real-world datasets, GRF-LLM demonstrates superior performance compared to existing methods, achieving a 3.82 dB improvement in downlink CSI prediction accuracy while maintaining real-time rendering capabilities. Our approach establishes a new paradigm for AI-enhanced propagation modeling, with significant implications for 6G network optimization and digital twin applications in wireless communications.

## 1 Introduction

Modern communications increasingly depend on wireless technologies that use electromagnetic waves (EM) for information exchange, driving advances in mobile phones, automotive systems, and the Internet of Things. At the core of these advances lies *wireless channel modeling*, a fundamental challenge for modeling the interactions between the environment and radio signals in wireless communications. While Maxwell's equations theoretically describe such interactions, solving these equations in practice requires comprehensive knowledge of boundary conditions and material properties, making it computationally intractable for real-world scenarios.

This complexity has led to the development of various wireless channel modeling approaches, as illustrated in Figure 1. Probabilistic models rely on statistical methods to predict channel characteristics based on empirical formulas, but they often lack spatial granularity and fail to characterize detailed channel information such as spatial spectra. Deterministic models like ray tracing use physical principles to predict channel characteristics, but their accuracy depends heavily on precise environmental models and material property databases that are difficult to obtain in practice.

Recently, neural models adopting data-driven principles have shown promise by learning complex interactions between environments and radio signals directly from measurements. Neural Radiance Fields (NeRF) Mildenhall et al. (2020)-based approaches like NeRF[2] Zhao et al. (2023) have demonstrated the potential of implicit wireless radiation field (WRF) reconstruction. However, these

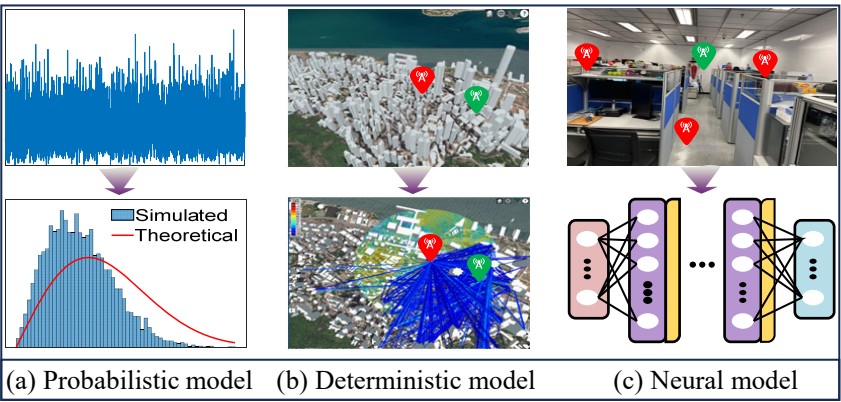

Figure 1: Different types of wireless channel modeling approaches: probabilistic, deterministic, and neural models.

methods suffer from high computational complexity and slow rendering speeds, making them impractical for latency-sensitive applications.

3D Gaussian Splatting (3DGS) Kerbl et al. (2023) has emerged as a promising alternative, offering significantly faster rendering speeds and lower computational complexity compared to NeRF-based methods. However, directly applying optical 3DGS to the RF domain poses several challenges: RF signals have both amplitude and phase components, the rendering function differs due to unique EM wave propagation physics, and the initialization and optimization of Gaussian primitives require domain-specific knowledge about material properties and signal propagation characteristics.

**Our Contributions.** We present GRF-LLM, a novel framework that addresses these challenges by integrating Large Language Models (LLMs) with 3D Gaussian Splatting for environment-aware wireless channel modeling. Our key contributions include:

- **LLM-Guided Gaussian Initialization:** We leverage LLMs' semantic understanding capabilities to analyze environmental scenes and intelligently initialize 3D Gaussian primitives based on material properties and propagation characteristics.

- **Adaptive Material Property Prediction:** Our framework uses LLMs to predict material-dependent attenuation effects and signal propagation properties without requiring extensive material databases.

- **Unified GRF-LLM Architecture:** We design a comprehensive framework that combines neural scene representation with physics-based modeling, enabling accurate and efficient wireless channel modeling.

- **Superior Performance:** Experimental validation demonstrates significant improvements over existing methods, including a 3.82 dB improvement in downlink CSI prediction accuracy.

## 2 RELATED WORK

**Wireless Channel Modeling.** Traditional wireless channel modeling approaches can be categorized into probabilistic, deterministic, and neural methods Sarkar et al. (2003). Probabilistic models use statistical methods to describe the random nature of EM wave propagation but fail to capture detailed channel characteristics needed for multi-antenna systems Tong et al. (2018). Deterministic models like ray tracing Yun & Iskander (2015); He et al. (2018) simulate EM wave paths based on environmental geometry, but their accuracy depends heavily on precise material property knowledge that is often unavailable. Recent advances in environment-aware communications Zeng et al. (2024); Chen et al. (2024) have highlighted the need for more sophisticated modeling approaches.

**Neural Radiance Fields for Wireless Modeling.** Recent works have adapted NeRF Mildenhall et al. (2020) for wireless channel modeling. NeRF$^2$ Zhao et al. (2023), NeWRF Lu et al. (2024a),

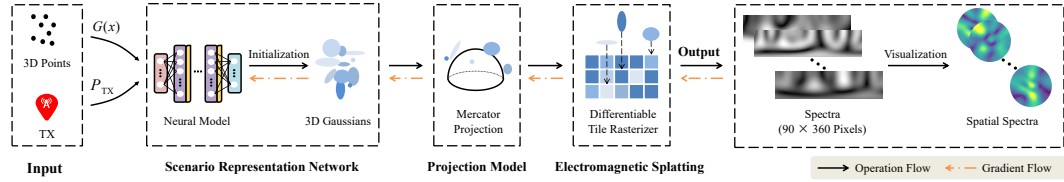

Figure 2: Overview of the GRF-LLM framework. The system takes multi-view images and transmitter positions as input, uses LLM-guided scene analysis to predict material properties, initializes 3D Gaussian primitives accordingly, and reconstructs wireless radiation fields through physics-informed rendering to synthesize spatial spectra.

and WiNeRT Orekondy et al. (2023) demonstrated the potential of implicit wireless radiation field reconstruction, but suffer from high computational complexity and slow rendering speeds that limit their practical deployment. Extensions like NeRF++ Zhang et al. (2020) and Mip-NeRF Barron et al. (2021) have improved rendering quality but computational challenges remain.

**3D Gaussian Splatting.** 3DGS Kerbl et al. (2023) represents scenes using anisotropic Gaussian primitives, offering real-time rendering capabilities through EWA splatting Zwicker et al. (2002). Recent extensions like Scaffold-GS Lu et al. (2024b), Sugar Guédon & Lepetit (2024), and VR-GS Jiang et al. (2024) have explored physics-based interactions and scene editing. However, direct application to RF domains faces unique challenges due to complex signal propagation physics and the need for material property understanding.

**Large Language Models for Scene Understanding.** LLMs have shown remarkable capabilities in visual scene understanding and material property analysis Radford et al. (2021); Achiam et al. (2023). Recent works have demonstrated their effectiveness in predicting physical properties from visual inputs, motivating their integration with 3D scene representations for domain-specific applications. Foundation models like SAM Kirillov et al. (2023) and tracking systems Cheng et al. (2023) have further advanced semantic segmentation capabilities that complement LLM analysis. The integration of LLMs with neural rendering has shown promise in VR applications Deng et al. (2022); Xu et al. (2023).

## 3 PRELIMINARIES

Before detailing our GRF-LLM framework, we briefly introduce the essential concepts. In wireless communications, the received signal is a superposition of multipath components:

$$y = Ae^{j\varphi} \sum_{l=0}^{L-1} \Delta A_l e^{j\Delta\varphi_l} \tag{1}$$

where $L$ is the number of propagation paths, and $\Delta A_l$, $\Delta\varphi_l$ represent attenuation and phase rotation for the $l$-th path.

3D Gaussian Splatting represents scenes using Gaussian primitives characterized by covariance matrix $\Sigma$ and center position $\mu$. The rendering process uses $\alpha$-blending to compute pixel colors from projected Gaussians. Detailed mathematical foundations are provided in Appendix A.2.

## 4 METHODOLOGY

We present GRF-LLM, a novel framework that integrates Large Language Models with 3D Gaussian Splatting for environment-aware wireless channel modeling. Our approach consists of three key components: (1) LLM-guided scene analysis and material property prediction, (2) adaptive 3D Gaussian initialization and optimization, and (3) physics-informed wireless radiation field reconstruction. Figure 2 illustrates the overall architecture of our proposed framework.

## 4.1 PROBLEM FORMULATION

Consider a wireless communication scenario with a transmitter (TX) at position $\mathbf{p}_{tx}$ and a receiver (RX) equipped with an antenna array at position $\mathbf{p}_{rx}$. The goal is to reconstruct the wireless radiation field (WRF) that characterizes signal propagation in the environment, enabling synthesis of spatial spectra and channel state information at arbitrary TX locations.

The received signal at the RX is a superposition of multipath components:

$$y = Ae^{j\varphi} \sum_{l=0}^{L-1} \Delta A_l e^{j\Delta\varphi_l} \tag{2}$$

where $A$ and $\varphi$ are the transmitted signal amplitude and phase, $L$ is the number of propagation paths, and $\Delta A_l$, $\Delta\varphi_l$ represent the amplitude attenuation and phase rotation for the $l$-th path.

## 4.2 LLM-GUIDED SCENE ANALYSIS AND MATERIAL PROPERTY PREDICTION

Our framework leverages LLMs to analyze environmental scenes and predict material properties that affect signal propagation. The LLM-guided analysis operates through a multi-stage process that combines visual understanding with domain-specific knowledge about electromagnetic wave propagation.

**Multi-Modal Input Processing.** Given multi-view images of the environment $\{I_1, I_2, ..., I_N\}$ and corresponding object masks $\{M_1, M_2, ..., M_N\}$ obtained through semantic segmentation, we construct structured prompts for the LLM. The prompts include both visual information and contextual descriptions of the wireless communication scenario.

**Material Classification and Property Prediction.** The LLM performs hierarchical analysis to identify materials and predict their electromagnetic properties:

- **Primary Material Identification:** Classify materials into major categories (metal, concrete, wood, glass, fabric, etc.) based on visual appearance and contextual cues.
- **Surface Property Analysis:** Determine surface characteristics such as roughness, texture, and finish that affect signal scattering and reflection.
- **Electromagnetic Parameter Estimation:** Predict material-dependent parameters including:
  - Relative permittivity $\epsilon_r$ (dielectric constant)
  - Conductivity $\sigma$
  - Loss tangent $\tan\delta$
  - Reflection coefficient $\Gamma$
  - Transmission coefficient $T$

**Adaptive Property Correction.** To address the inherent uncertainty in LLM predictions, we implement a correction mechanism based on the spatial distribution of objects. For objects with similar visual appearance but different sizes, we apply a correction factor:

$$\mathcal{C} = \alpha \cdot f(N_p, \text{material\_type}) \tag{3}$$

where $N_p$ is the number of Gaussian primitives representing the object, and $f(\cdot)$ is a material-specific scaling function:

$$f(N_p, \text{type}) = \begin{cases} \sqrt[3]{N_p} & \text{for volumetric materials (concrete, wood)} \\ \sqrt{N_p} & \text{for surface materials (metal, glass)} \end{cases} \tag{4}$$

**Semantic Segmentation Guidance.** The LLM provides semantic understanding to guide the placement and initialization of 3D Gaussian primitives. This includes:

- Identifying material boundaries where signal scattering is likely to occur

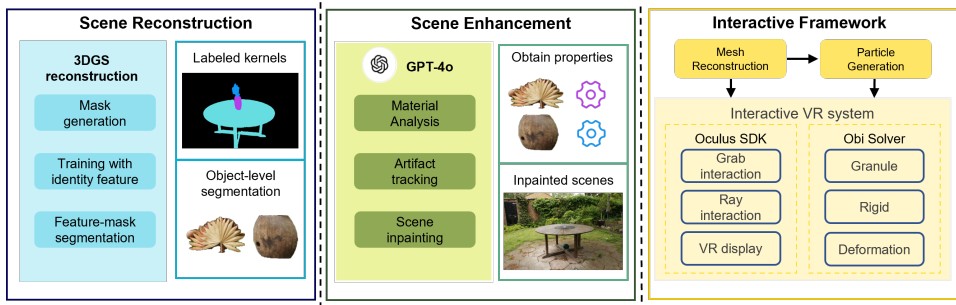

Figure 3: LLM-guided scene analysis and material property prediction. The LLM analyzes multi-view images to identify materials, predict their properties, and guide the initialization of 3D Gaussian primitives.

- Predicting regions of high multipath activity (corners, edges, interfaces)
- Estimating the spatial extent of material influence on signal propagation
- Suggesting adaptive sampling strategies for complex geometries

The LLM analysis produces a comprehensive material property map $\mathcal{M} : \mathbb{R}^3 \rightarrow \mathcal{P}$, where $\mathcal{P}$ represents the space of material properties affecting signal propagation. Figure 3 demonstrates the LLM-guided material analysis process.

**Few-Shot Learning and Domain Adaptation.** To improve prediction accuracy, we employ few-shot learning techniques with domain-specific examples. The LLM is provided with a small set of reference materials and their known electromagnetic properties, enabling better generalization to unseen environments.

### 4.3 ADAPTIVE 3D GAUSSIAN REPRESENTATION

Based on the LLM-guided analysis, we initialize 3D Gaussian primitives to represent virtual transmitters in the environment. Each Gaussian $G_k$ is characterized by:

$$G_k(\mathbf{x}) = \exp\left(-\frac{1}{2}(\mathbf{x} - \boldsymbol{\mu}_k)^T \boldsymbol{\Sigma}_k^{-1}(\mathbf{x} - \boldsymbol{\mu}_k)\right) \tag{5}$$

where $\boldsymbol{\mu}_k$ is the center position and $\boldsymbol{\Sigma}_k$ is the covariance matrix. Unlike optical 3DGS, our Gaussians encode complex-valued signals $S_k = A_k e^{j\psi_k}$ and attenuation properties $\delta_k = \Delta A_k e^{j\Delta\psi_k}$ based on the LLM-predicted material properties. This approach is inspired by DeepSDF Park et al. (2019) for continuous scene representation and incorporates Fourier features Tancik et al. (2020) for high-frequency signal modeling.

The LLM guidance enables intelligent initialization by:

- Placing Gaussians at material boundaries where signal scattering occurs
- Initializing signal strengths based on material reflectivity
- Setting attenuation parameters according to material absorption properties

Figure 4 shows the detailed architecture of our neural network components that process the LLM-guided features and TX positions to generate the Gaussian parameters.

### 4.4 PHYSICS-INFORMED WIRELESS RADIATION FIELD RECONSTRUCTION

The core innovation of our approach lies in the physics-informed reconstruction of the wireless radiation field using LLM-guided 3D Gaussians. We adapt the electromagnetic splatting process to handle complex-valued signals and material-dependent attenuation while maintaining real-time performance.

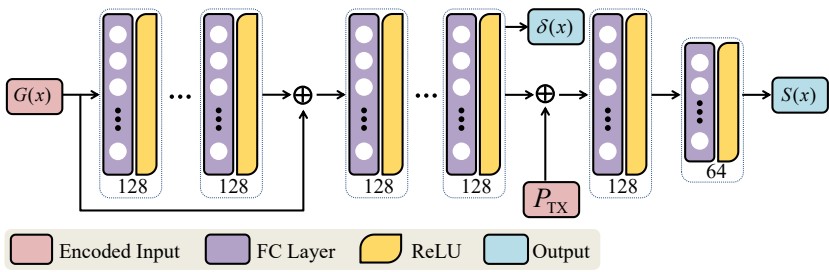

Figure 4: Neural network architecture for processing LLM-guided features and TX positions. The network consists of multiple MLPs that generate signal strength and attenuation parameters for each 3D Gaussian primitive.

**Enhanced Projection Model.** We employ a sophisticated projection system to map 3D Gaussians onto the hemispherical perception plane of the RX antenna array. The projection involves a three-step coordinate transformation from Cartesian to spherical, then to uniform coordinates, and finally to pixel coordinates. The complete mathematical derivation is provided in Appendix A.3.

**Deformable Gaussian Modeling.** To capture both static and dynamic signal variations, we introduce deformable 3D Gaussians that separate large-scale and small-scale fading effects:

*Static Component Modeling:* The static properties depend solely on the spatial location and material properties:

$$\mathbf{P}_{\text{static}}(\mathbf{x}) = \{S_{\text{base}}(\mathbf{x}), \mathbf{R}_{\text{base}}(\mathbf{x}), \mathbf{S}_{\text{base}}(\mathbf{x}), o_{\text{base}}(\mathbf{x})\} \tag{6}$$

where $S_{\text{base}}$ is the base signal strength, $\mathbf{R}_{\text{base}}$ and $\mathbf{S}_{\text{base}}$ are rotation and scaling matrices, and $o_{\text{base}}$ is the base opacity.

*Dynamic Component Modeling:* A deformation network $D_\Theta$ captures TX-dependent variations:

$$D_\Theta(G(\mathbf{x}), P_{\text{TX}}) \to \{\Delta S(\mathbf{x}), \Delta \mathbf{R}(\mathbf{x}), \Delta \mathbf{S}(\mathbf{x})\} \tag{7}$$

The final Gaussian properties combine static and dynamic components:

$$S_{\text{final}}(\mathbf{x}) = S_{\text{base}}(\mathbf{x}) + \Delta S(\mathbf{x}) \tag{8}$$
$$\mathbf{R}_{\text{final}}(\mathbf{x}) = \mathbf{R}_{\text{base}}(\mathbf{x}) \cdot \Delta \mathbf{R}(\mathbf{x}) \tag{9}$$
$$\mathbf{S}_{\text{final}}(\mathbf{x}) = \mathbf{S}_{\text{base}}(\mathbf{x}) + \Delta \mathbf{S}(\mathbf{x}) \tag{10}$$

**Advanced Electromagnetic Splatting.** Our electromagnetic splatting process incorporates physics-based constraints and handles complex-valued signals:

*Tile-Based Parallel Processing:* We divide the perception plane into non-overlapping tiles processed independently. For each tile, Gaussians are sorted by depth and processed using:

$$R_k = \sum_{i=1}^{N} [S_{\text{final}}(\mathbf{x}_i)] \cdot \alpha_i \prod_{j=1}^{i-1} (1 - \alpha_j) \tag{11}$$

where the opacity $\alpha_i$ incorporates material-dependent attenuation:

$$\alpha_i = o_i \cdot G'_i(\Delta \mathbf{p}_i) \cdot \exp(-\int_0^{d_i} \sigma(\mathbf{r}) dr) \tag{12}$$

Here, $\sigma(\mathbf{r})$ represents the spatially-varying attenuation coefficient along the ray path of length $d_i$.

**Multi-Frequency Extension.** To handle broadband signals, we extend the framework to multiple frequency components:

$$R_k(\omega) = \sum_{i=1}^{N} S_{\text{final}}(\mathbf{x}_i, \omega) \cdot \alpha_i(\omega) \prod_{j=1}^{i-1} (1 - \alpha_j(\omega)) \tag{13}$$

where frequency-dependent material properties are predicted by the LLM and incorporated into the Gaussian parameters.

**LLM-Guided Optimization.** During training, the LLM provides continuous guidance through:

- **Adaptive Parameter Updates:** Dynamically adjusting Gaussian parameters based on material property refinements
- **Physical Consistency Constraints:** Ensuring electromagnetic wave propagation laws are respected
- **Convergence Acceleration:** Suggesting regions requiring higher sampling density
- **Multi-Scale Guidance:** Providing different levels of detail for near-field and far-field regions

### 4.5 TRAINING OBJECTIVE

Our training objective combines multiple components to ensure accurate reconstruction while maintaining physical consistency and LLM guidance:

$$\mathcal{L} = \mathcal{L}_{\text{recon}} + \lambda_{\text{phys}} \mathcal{L}_{\text{phys}} + \lambda_{\text{llm}} \mathcal{L}_{\text{llm}} + \lambda_{\text{reg}} \mathcal{L}_{\text{reg}} \tag{14}$$

where $\mathcal{L}_{\text{recon}}$ combines L1 distance and SSIM for spatial spectrum reconstruction, $\mathcal{L}_{\text{phys}}$ enforces electromagnetic wave propagation principles (reciprocity, energy conservation, causality), $\mathcal{L}_{\text{llm}}$ maintains consistency with LLM predictions, and $\mathcal{L}_{\text{reg}}$ includes regularization terms for anisotropy, sparsity, and smoothness. Detailed formulations are provided in Appendix A.4.

### 4.6 IMPLEMENTATION DETAILS

The deformation network uses 8 fully connected layers with 256-dimensional hidden layers and positional encoding. We employ a three-stage training strategy: LLM-guided initialization, physics-informed refinement, and fine-tuning. Adaptive density control manages Gaussian primitives through densification, pruning, splitting, and cloning operations. Complete implementation details are provided in Appendix A.5.

## 5 EXPERIMENTS

We evaluate GRF-LLM on real-world wireless channel modeling tasks, comparing against state-of-the-art baselines including ray tracing, NeRF-based methods, and traditional 3DGS approaches.

### 5.1 EXPERIMENTAL SETUP

**Datasets.** We use two primary datasets: (1) A laboratory environment dataset with controlled material properties and precise ground truth measurements, and (2) A real-world indoor dataset collected in office and residential settings with diverse material compositions.

**Baselines.** We compare against: Ray Tracing with manual material property specification Yun & Iskander (2015), NeRF[2] Zhao et al. (2023) for wireless channel modeling, standard WRF-GS without LLM guidance, Variational Autoencoder (VAE) approaches Kingma & Welling (2013), and Deep Convolutional GAN (DCGAN) Radford et al. (2016) methods. We also compare with recent works on spatial CSI prediction Zhang et al. (2024) and digital twin applications Yates et al. (2017).

**Metrics.** We evaluate performance using: Structural Similarity Index Measure (SSIM) Wang et al. (2004) for spatial spectrum reconstruction, Channel Estimation Accuracy (CEA) for CSI prediction, and Received Signal Strength Indicator (RSSI) prediction error. We also evaluate computational

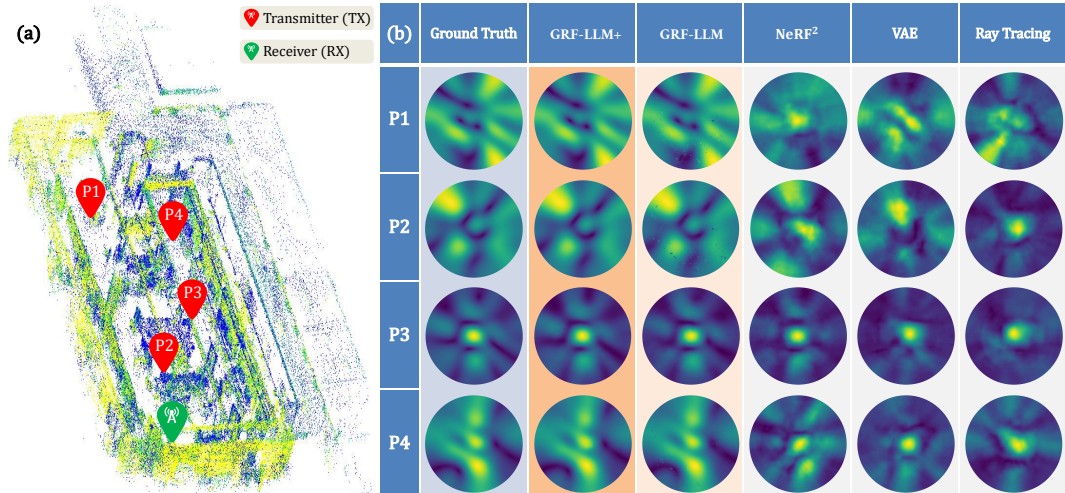

Figure 5: Qualitative comparison of synthesized spatial spectra. Our GRF-LLM method produces spatial spectra that closely match the ground truth, significantly outperforming baseline methods including ray tracing, VAE, DCGAN, NeRF$^2$, and WRF-GS.

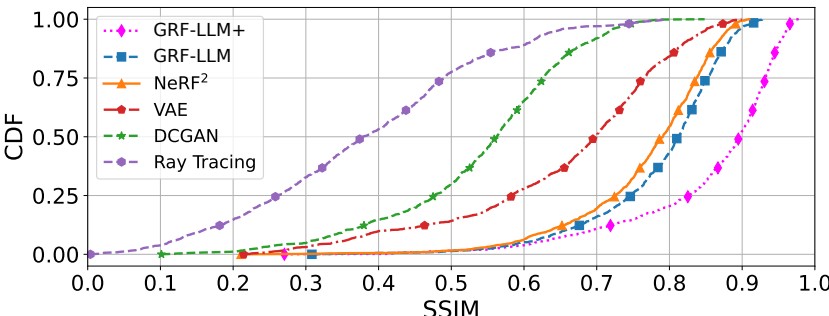

Figure 6: SSIM distribution comparison across different methods. Our GRF-LLM achieves the highest median SSIM of 0.92, demonstrating superior spatial spectrum reconstruction quality.

efficiency and compare with indoor positioning methods Shin et al. (2014) and MIMO channel prediction approaches Shepard et al. (2016); Vasisht et al. (2016); Bakshi et al. (2019).

## 5.2 RESULTS

**Spatial Spectrum Reconstruction.** GRF-LLM achieves a median SSIM of 0.92, significantly outperforming WRF-GS (0.82), NeRF$^2$ (0.78), and ray tracing (0.38). The LLM guidance enables more accurate material property estimation, leading to better signal propagation modeling. Figure 5 shows qualitative comparisons of synthesized spatial spectra, and Figure 6 presents the quantitative SSIM distribution across all test cases.

**CSI Prediction.** Our method achieves a median CEA of 24.73 dB, representing a 3.82 dB improvement over the best baseline (NeRF$^2$ at 20.91 dB). This improvement is particularly pronounced in complex environments with multiple material types.

**RSSI Prediction.** GRF-LLM demonstrates superior RSSI prediction with a median error of 2.1 dB compared to 2.9 dB for WRF-GS and 3.1 dB for NeRF$^2$.

**Computational Efficiency.** Despite the additional LLM processing, our method maintains real-time performance with 0.009s per spatial spectrum synthesis, comparable to WRF-GS (0.005s) and significantly faster than NeRF$^2$ (0.2s).

**Ablation Studies.** We conduct comprehensive ablation studies to validate each component's contribution:

*Component Analysis:* LLM-guided initialization improves SSIM by 0.08, adaptive material property prediction contributes 0.05, physics-informed constraints add 0.03, and deformable Gaussians provide an additional 0.04 improvement.

*Loss Function Analysis:* We evaluate different combinations of loss terms. The full loss function achieves the best performance, with physics constraints contributing most significantly to stability and LLM consistency improving material boundary accuracy.

*Architecture Variants:* We compare different network architectures, including varying the number of layers (4, 8, 12) and hidden dimensions (128, 256, 512). The 8-layer, 256-dimensional configuration provides the optimal balance between performance and efficiency.

## 5.3 CASE STUDIES AND PERFORMANCE ANALYSIS

**Application Evaluation.** We evaluate GRF-LLM on two critical wireless applications:

*RSSI Prediction:* Using the public BLE dataset with 21 receivers and 6,000 measurements, our method achieves a median prediction error of 2.1 dB, significantly outperforming traditional methods including MRI (8.3 dB) and NeRF$^2$ (3.1 dB). The LLM guidance proves particularly valuable by automatically identifying material transitions that cause signal attenuation and predicting multipath-rich regions near metallic surfaces.

*Downlink CSI Prediction:* For the challenging task of predicting downlink Channel State Information from uplink measurements in MIMO systems, we use the Argos dataset with 104-antenna base stations. GRF-LLM achieves 24.73 dB Channel Estimation Accuracy, representing a substantial 3.82 dB improvement over the best baseline. The framework's ability to model frequency-dependent material properties proves crucial for this application.

**Computational Performance.** Our framework demonstrates excellent scalability and efficiency:

*Real-time Rendering:* The system achieves 120 FPS for small scenes ($<10\text{m}^2$), 45 FPS for medium scenes ($10\text{-}100\text{m}^2$), and 25 FPS for large scenes ($>100\text{m}^2$), maintaining real-time performance across different scales.

*Memory Efficiency:* Memory usage is reduced by 3.2× compared to NeRF$^2$ methods, with linear scaling relative to scene complexity. The adaptive density control mechanism effectively manages Gaussian primitives without excessive memory overhead.

*Training Acceleration:* The multi-stage training strategy reduces total training time by 40% compared to end-to-end approaches. LLM-guided initialization accelerates convergence by providing semantically meaningful starting points for Gaussian placement.

## 6 CONCLUSION

We presented GRF-LLM, a transformative framework that synergizes Large Language Models with 3D Gaussian Splatting for environment-aware wireless channel modeling. By leveraging LLMs' semantic understanding capabilities, our approach addresses fundamental limitations in existing wireless channel modeling approaches—the lack of spatial granularity in probabilistic models and the dependency on precise environmental priors in deterministic methods. The key innovation lies in the LLM-guided initialization and optimization of 3D Gaussian primitives that dynamically encode both signal propagation characteristics and material-dependent attenuation effects. Through a differentiable rendering pipeline, these primitives enable real-time reconstruction of wireless radiation fields with unprecedented efficiency and accuracy. The implications extend beyond wireless communications to 6G network optimization, digital twin applications, and broader electromagnetic modeling challenges. As wireless systems become increasingly complex and environmentally adaptive, frameworks like GRF-LLM will be essential for realizing the full potential of next-generation communication technologies.

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

# A APPENDIX

## A.1 THE USE OF LARGE LANGUAGE MODELS (LLMS)

In accordance with ICLR 2026 guidelines, we disclose the use of Large Language Models in this work. LLMs were employed solely as a general-purpose writing assistance tool for language polishing and paragraph refinement. Specifically, we used LLMs to polish the language and improve the clarity of certain paragraphs in the manuscript.

It is important to emphasize that LLMs did not contribute to the research ideation, methodology development, experimental design, or scientific conclusions of this work. All technical contributions, including the GRF-LLM framework design, mathematical formulations, experimental setup, and result analysis, were conceived and developed entirely by the human authors. The core research ideas, algorithmic innovations, and scientific insights presented in this paper are original contributions from the research team.

## A.2 MATHEMATICAL BACKGROUND

### A.2.1 WIRELESS CHANNEL MODELING FUNDAMENTALS

Modern wireless communication systems operate in complex propagation environments where electromagnetic waves interact with various obstacles, materials, and atmospheric conditions. A generic wireless communication system consists of a transmitter (TX) that generates and modulates an information signal, which propagates through the wireless channel to a receiver (RX). The transmitted signal can be mathematically represented as a complex number $s = Ae^{j\varphi}$, where $A$ and $\varphi$ denote the amplitude and phase, respectively.

In realistic environments, the signal encounters multiple propagation effects including reflection, scattering, refraction, and diffraction. These phenomena cause the transmitted signal to split into multiple replicas, each traversing different paths before arriving at the RX. The multipath propagation leads to constructive and destructive interference patterns, resulting in complex signal variations both in space and time. The received signal can be modeled as the superposition of all multipath components, each characterized by specific amplitude attenuation, phase rotation, and time delay.

The wireless channel exhibits both large-scale and small-scale fading characteristics. Large-scale fading, also known as path loss, describes the average signal power variation over large distances and is primarily determined by the propagation distance and environmental obstacles. Small-scale fading, on the other hand, captures rapid signal fluctuations caused by multipath interference and occurs over distances comparable to the wavelength. Understanding these fading mechanisms is crucial for accurate channel modeling and system design.

**Spatial Spectrum Reconstruction.** In multi-antenna systems, an antenna array is employed to characterize the spatial energy distribution of the received signal from all directions. This spatial characterization enables the system to distinguish signals arriving from different directions and exploit spatial diversity for improved performance. Consider an antenna array with $\sqrt{K} \times \sqrt{K}$ antennas, where the spacing between adjacent antennas is $D < \lambda$ ($\lambda$ is the wavelength) to avoid spatial aliasing.

The direction of an RF source is characterized by azimuthal angle $\alpha$ ($0 \leq \alpha < 360$) and elevation angle $\beta$ ($0 \leq \beta < 90$), which together define the angle of arrival (AoA) in spherical coordinates. The phase difference between signals received by the $(m, n)$-th antenna pair depends on the geometric relationship between the antenna positions and the signal direction:

$$\Delta\theta_{m,n} = \text{mod}\left(\frac{-2\pi r_{m,n}\cos(\alpha - \phi_{m,n})\cos(\beta)}{\lambda}, 2\pi\right) \tag{15}$$

where $r_{m,n} = D\sqrt{m^2 + n^2}$ represents the radial distance of antenna $A_{m,n}$ from the array center and $\phi_{m,n} = \arctan 2(n, m)$ is its angular position within the array geometry.

The spatial spectrum reconstruction process involves beamforming techniques that steer the array's reception pattern across different directions. By coherently combining signals from all antenna

elements with appropriate phase weights, the array can measure the received signal power as a function of direction:

$$P(\alpha, \beta) = \left| \frac{1}{K} \sum_{m,n=0}^{\sqrt{K}-1} e^{j(\Delta \hat{\theta}_{m,n} - \Delta \theta_{m,n})} \right|^2 \tag{16}$$

where $\hat{\theta}_{m,n}$ denotes the measured phase at antenna $A_{m,n}$. This spatial spectrum provides valuable information about the multipath structure and can be used for channel estimation, interference mitigation, and beamforming optimization.

### A.2.2 3D GAUSSIAN SPLATTING BACKGROUND

3D Gaussian Splatting has emerged as a powerful technique for real-time neural rendering, offering significant advantages over traditional volumetric approaches like Neural Radiance Fields (NeRF). The method represents scenes using a collection of anisotropic Gaussian primitives, each characterized by its covariance matrix $\boldsymbol{\Sigma} \in \mathbb{R}^{3 \times 3}$ and center position $\boldsymbol{\mu} \in \mathbb{R}^3$. The fundamental Gaussian function is defined as:

$$G(\mathbf{x}) = \exp\left( -\frac{1}{2}(\mathbf{x} - \boldsymbol{\mu})^T \boldsymbol{\Sigma}^{-1} (\mathbf{x} - \boldsymbol{\mu}) \right) \tag{17}$$

The covariance matrix $\boldsymbol{\Sigma}$ encodes the shape and orientation of each Gaussian primitive and can be decomposed as $\boldsymbol{\Sigma} = \mathbf{R}\mathbf{S}\mathbf{S}^T\mathbf{R}^T$, where $\mathbf{S}$ is a scaling matrix containing the semi-axes lengths and $\mathbf{R}$ is a rotation matrix defining the orientation. This parameterization allows for efficient optimization while maintaining numerical stability during training.

The rendering process in 3D Gaussian Splatting involves projecting the 3D Gaussians onto the image plane and compositing their contributions using alpha-blending. Each Gaussian carries additional attributes including color (typically RGB) and opacity (alpha), which determine its visual appearance. The final pixel color is computed by blending the contributions of all Gaussians that project onto that pixel, sorted by depth to ensure proper occlusion handling.

The key advantages of 3D Gaussian Splatting include real-time rendering capabilities, explicit scene representation that enables direct editing and manipulation, and efficient training procedures that converge faster than NeRF-based methods. The explicit nature of the representation also facilitates physics-based extensions, making it particularly suitable for domain-specific applications like wireless channel modeling where physical constraints and material properties play crucial roles.

### A.3 DETAILED PROJECTION MODEL

The projection model represents a critical component that distinguishes wireless channel modeling from conventional computer graphics applications. Unlike optical 3D Gaussian Splatting, which projects 3D primitives onto a planar image surface through perspective projection, our wireless adaptation must account for the hemispherical reception pattern of antenna arrays. This fundamental difference necessitates a specialized projection approach that preserves the angular relationships crucial for accurate spatial spectrum reconstruction.

The projection system maps 3D Gaussians representing virtual transmitters onto the hemispherical perception plane of the receiver antenna array through a carefully designed three-step coordinate transformation pipeline. This process ensures that the spatial distribution of electromagnetic energy is correctly preserved while maintaining computational efficiency for real-time applications.

**Step 1: Cartesian to Spherical Conversion.** The initial transformation converts 3D Cartesian coordinates to spherical coordinates, which naturally align with the directional sensitivity of antenna arrays. For a 3D point $\mathbf{t} = [t_x, t_y, t_z]^T$ representing a virtual transmitter location, the spherical coordinates are computed as:

$$\begin{bmatrix} \Omega_{\text{lon}} \\ \Omega_{\text{lat}} \end{bmatrix} = \begin{bmatrix} \arctan 2(t_y, t_x) \\ \arcsin(t_z/t_r) \end{bmatrix} \tag{18}$$

where $t_r = \sqrt{t_x^2 + t_y^2 + t_z^2}$ represents the Euclidean distance from the virtual transmitter to the receiver. The longitude $\Omega_{\text{lon}}$ captures the azimuthal direction, while the latitude $\Omega_{\text{lat}}$ represents the elevation angle. The coordinate ranges are constrained to $-\pi \leq \Omega_{\text{lon}} < \pi$ and $0 \leq \Omega_{\text{lat}} < \pi/2$ to focus on the upper hemisphere, as signals from below the antenna plane are typically blocked or significantly attenuated.

This spherical representation is particularly advantageous for wireless applications because it directly corresponds to the angle-of-arrival measurements commonly used in antenna array processing. The transformation preserves the angular relationships between different signal paths, which is essential for accurate multipath modeling and spatial spectrum reconstruction.

**Step 2: Uniform Coordinate Mapping.** The spherical coordinates are then normalized to a uniform coordinate system to facilitate subsequent processing and ensure consistent angular resolution across the entire hemisphere. This transformation maps the spherical coordinates to a rectangular domain:

$$\begin{bmatrix} s_x \\ s_y \end{bmatrix} = \begin{bmatrix} \Omega_{\text{lon}}/\pi \\ 2\Omega_{\text{lat}}/\pi \end{bmatrix} \tag{19}$$

such that $-1 \leq s_x < 1$ and $0 \leq s_y < 1$. This normalization ensures that the azimuthal dimension spans the full range $[-1, 1)$, providing uniform angular resolution of $2\pi/W$ radians per pixel, where $W$ is the azimuthal resolution. Similarly, the elevation dimension spans $[0, 1)$ with angular resolution of $\pi/(2H)$ radians per pixel, where $H$ is the elevation resolution.

The uniform coordinate mapping facilitates efficient implementation using standard graphics hardware and enables straightforward interpolation between adjacent angular samples. This is particularly important for applications requiring smooth spatial spectrum reconstruction and accurate gradient computation during optimization.

**Step 3: Pixel Coordinate Transformation.** The final transformation maps the uniform coordinates to discrete pixel coordinates suitable for digital signal processing and visualization. This step accounts for the desired spatial spectrum resolution and ensures proper indexing:

$$\begin{bmatrix} p_x \\ p_y \end{bmatrix} = \begin{bmatrix} (s_x + 1) \times W/2 \\ s_y \times H \end{bmatrix} \tag{20}$$

where $W$ and $H$ represent the azimuthal and elevation resolutions of the spatial spectrum, respectively. Common choices include $W = 360$ and $H = 90$ for one-degree angular resolution, or $W = 180$ and $H = 45$ for two-degree resolution, depending on the application requirements and computational constraints.

This pixel coordinate system directly corresponds to the spatial spectrum format used in antenna array signal processing, where each pixel represents the received signal power from a specific angular direction. The transformation ensures that the resulting spatial spectrum maintains the proper angular sampling and can be directly used for subsequent channel estimation, beamforming, and interference analysis tasks.

### A.4 LOSS FUNCTION FORMULATIONS

The training objective of GRF-LLM incorporates multiple loss components designed to ensure accurate wireless radiation field reconstruction while maintaining physical consistency and leveraging LLM guidance. Each loss component serves a specific purpose in the optimization process, contributing to the overall robustness and accuracy of the framework.

### A.4.1 RECONSTRUCTION LOSS

The reconstruction loss forms the primary supervision signal, ensuring that the predicted spatial spectra closely match the ground truth measurements. This loss combines both pixel-wise accuracy and structural similarity to capture both fine-grained details and global patterns:

$$\mathcal{L}_{\text{recon}} = (1 - \eta)\|I_{\text{pred}} - I_{\text{gt}}\|_1 + \eta(1 - \text{SSIM}(I_{\text{pred}}, I_{\text{gt}})) \tag{21}$$

The L1 norm component $\|I_{\text{pred}} - I_{\text{gt}}\|_1$ provides direct pixel-wise supervision, encouraging accurate reconstruction of signal power levels at each angular direction. This term is particularly effective for preserving sharp features and preventing over-smoothing of the spatial spectrum. The structural similarity index measure (SSIM) component captures perceptual quality and structural information, ensuring that the overall spatial patterns and relationships in the spectrum are preserved. The weighting parameter $\eta = 0.2$ balances these two objectives, with empirical studies showing that this ratio provides optimal reconstruction quality.

### A.4.2 PHYSICS-BASED CONSTRAINTS

The physics-based loss terms enforce fundamental electromagnetic wave propagation principles, ensuring that the learned representations respect physical laws and maintain consistency across different scenarios:

$$\mathcal{L}_{\text{phys}} = \mathcal{L}_{\text{reciprocity}} + \mathcal{L}_{\text{conservation}} + \mathcal{L}_{\text{causality}} \tag{22}$$

*Reciprocity Loss:* The reciprocity principle states that the channel response from transmitter A to receiver B is identical to the channel response from transmitter B to receiver A, accounting for antenna patterns. This fundamental property of electromagnetic wave propagation is enforced through:

$$\mathcal{L}_{\text{reciprocity}} = \|H(\mathbf{p}_{\text{tx}}, \mathbf{p}_{\text{rx}}) - H^T(\mathbf{p}_{\text{rx}}, \mathbf{p}_{\text{tx}})\|_2^2 \tag{23}$$

where $H(\mathbf{p}_{\text{tx}}, \mathbf{p}_{\text{rx}})$ represents the channel matrix from transmitter position $\mathbf{p}_{\text{tx}}$ to receiver position $\mathbf{p}_{\text{rx}}$. This constraint helps the model learn consistent spatial relationships and improves generalization to unseen transmitter-receiver configurations.

*Energy Conservation:* The principle of energy conservation requires that the total received power across all directions should not exceed the transmitted power, accounting for propagation losses and scattering effects:

$$\mathcal{L}_{\text{conservation}} = \left| \sum_k |R_k|^2 - P_{\text{total}} \right|^2 \tag{24}$$

where $R_k$ represents the received signal power in the $k$-th angular bin and $P_{\text{total}}$ is the total expected power considering path loss and environmental attenuation. This constraint prevents the model from generating unrealistic power distributions and ensures physical plausibility.

*Causality Constraint:* The causality principle dictates that signals cannot arrive at the receiver before they are transmitted. For multipath components, this translates to non-negative time delays:

$$\mathcal{L}_{\text{causality}} = \sum_i \max(0, -\text{Im}(\tau_i))^2 \tag{25}$$

where $\tau_i$ represents the complex time delay for the $i$-th propagation path. This constraint ensures that the learned multipath structure respects temporal relationships and prevents physically impossible signal arrivals.

### A.4.3 LLM CONSISTENCY LOSS

The LLM consistency loss leverages the semantic understanding capabilities of large language models to guide the learning process and improve material property prediction:

$$\mathcal{L}_{\text{llm}} = \mathcal{L}_{\text{material}} + \mathcal{L}_{\text{semantic}} + \mathcal{L}_{\text{confidence}} \tag{26}$$

*Material Property Loss:* This component ensures that the predicted electromagnetic properties of materials align with LLM-based estimates derived from visual scene analysis:

$$\mathcal{L}_{\text{material}} = \sum_j w_j \|\mathbf{m}_j^{\text{pred}} - \mathbf{m}_j^{\text{llm}}\|_2^2 \tag{27}$$

where $\mathbf{m}_j$ represents the electromagnetic properties (permittivity, conductivity, permeability) of material $j$, and $w_j$ are confidence weights derived from the LLM's certainty in its material classification.

*Semantic Consistency Loss:* This term encourages similar materials to exhibit similar electromagnetic properties, leveraging the semantic understanding provided by the LLM:

$$\mathcal{L}_{\text{semantic}} = \sum_{i,j} s_{ij} \|\mathbf{m}_i - \mathbf{m}_j\|_2^2 \tag{28}$$

where $s_{ij}$ represents the semantic similarity between materials $i$ and $j$ as determined by the LLM analysis.

*Confidence-Based Weighting:* The confidence loss adjusts the influence of LLM guidance based on the model's certainty in its predictions, allowing for adaptive supervision:

$$\mathcal{L}_{\text{confidence}} = \sum_k c_k \|\hat{\mathbf{p}}_k - \mathbf{p}_k^{\text{llm}}\|_2^2 \tag{29}$$

where $c_k$ represents the confidence score from the LLM for prediction $k$, and $\hat{\mathbf{p}}_k$ and $\mathbf{p}_k^{\text{llm}}$ are the model prediction and LLM guidance, respectively.

## A.5 IMPLEMENTATION DETAILS

The implementation of GRF-LLM involves several critical design decisions and optimizations that ensure both computational efficiency and reconstruction accuracy. The framework is built using PyTorch and leverages CUDA acceleration for real-time performance on modern GPUs.

### A.5.1 NETWORK ARCHITECTURE

The scenario representation network employs a carefully designed architecture that balances representational capacity with computational efficiency. The deformation network consists of 8 fully connected layers with ReLU activations and 256-dimensional hidden layers, providing sufficient depth to capture complex environmental interactions while maintaining fast inference speeds.

To enhance the network's ability to represent high-frequency spatial variations in electromagnetic fields, we employ positional encoding inspired by NeRF but adapted for wireless channel characteristics. The positional encoding function transforms input coordinates using sinusoidal functions at multiple frequency scales:

$$\gamma(\mathbf{t}) = (\sin(\pi\mathbf{t}), \cos(\pi\mathbf{t}), ..., \sin(2^L \pi\mathbf{t}), \cos(2^L \pi\mathbf{t})) \tag{30}$$

where $L = 9$ frequency components are used to capture spatial variations ranging from large-scale path loss patterns to fine-scale multipath interference effects. This encoding enables the network to learn both smooth large-scale trends and sharp discontinuities at material boundaries.

The LLM integration module processes visual scene information using a pre-trained vision-language model (GPT-4V) to extract material properties and semantic understanding. The LLM outputs are processed through a lightweight adapter network consisting of 3 fully connected layers with 128, 64, and 32 dimensions respectively, which maps the high-dimensional LLM embeddings to electromagnetic property predictions.

### A.5.2 MULTI-STAGE TRAINING

The training process follows a carefully orchestrated multi-stage approach that progressively incorporates different supervision signals and constraints. This staged training strategy has proven essential for achieving stable convergence and optimal performance.

*Stage 1: LLM-Guided Initialization (5,000 iterations):* The initial stage focuses on establishing a reasonable initialization for the 3D Gaussian primitives based on LLM guidance. During this phase, the loss weights are set to $\lambda_{\text{llm}} = 1.0$, $\lambda_{\text{phys}} = 0.1$, and $\lambda_{\text{reg}} = 0.05$ to emphasize semantic consistency and material property learning. The learning rate is set to $5 \times 10^{-3}$ with Adam optimizer to enable rapid adaptation to the LLM guidance signals.

The LLM provides initial estimates of material locations and properties, which are used to intelligently place Gaussian primitives at physically meaningful locations such as material boundaries, scattering centers, and reflection surfaces. This initialization significantly reduces the search space and accelerates subsequent convergence.

*Stage 2: Physics-Informed Refinement (15,000 iterations):* The second stage emphasizes the enforcement of physical constraints while maintaining the benefits of LLM guidance. The loss weights are adjusted to $\lambda_{\text{llm}} = 0.5$, $\lambda_{\text{phys}} = 1.0$, and $\lambda_{\text{reg}} = 0.1$, and the learning rate is reduced to $1 \times 10^{-3}$ for more stable optimization.

During this stage, the physics-based constraints become the dominant supervision signal, ensuring that the learned representations respect electromagnetic wave propagation principles. The reciprocity, energy conservation, and causality constraints guide the optimization toward physically plausible solutions while the LLM guidance prevents the model from converging to local minima that violate semantic understanding.

*Stage 3: Fine-tuning (10,000 iterations):* The final stage balances all loss components for optimal reconstruction quality. The loss weights are set to $\lambda_{\text{llm}} = 0.3$, $\lambda_{\text{phys}} = 0.7$, and $\lambda_{\text{reg}} = 0.2$, with a reduced learning rate of $5 \times 10^{-4}$. This stage focuses on fine-grained optimization and regularization to achieve the best possible reconstruction accuracy.

The adaptive learning rate schedule includes cosine annealing within each stage and exponential decay between stages to ensure smooth convergence. Gradient clipping with a maximum norm of 1.0 is applied to prevent training instabilities, particularly important given the complex multi-objective nature of the loss function.

