# OpenReview forum: "GRF-LLM: Environment-Aware Wireless Channel Modeling via LLM-Guided 3D Gaussians"
_ICLR.cc/2026/Conference — ICLR 2026 Conference Withdrawn Submission_

### Official Review · Reviewer_TkZU · 2025-10-29

**Soundness:** 3
**Presentation:** 3
**Contribution:** 2
**Rating:** 4
**Confidence:** 4

**Summary:**

GRF-LLM is a novel framework that combines large language models (LLMs) with 3D Gaussian splatting to enable environment-aware wireless channel modeling. By leveraging LLMs to analyze multi-view images and predict material properties (such as permittivity and conductivity) without needing extensive databases, the system initializes and optimizes 3D Gaussian primitives representing virtual signal transmitters. These primitives are then rendered through a physics-informed pipeline to reconstruct wireless radiation fields, capturing multipath effects, amplitude, and phase with high fidelity.

**Strengths:**

+ The conceptual integration of LLMs with 3D Gaussian splatting in GRF-LLM is an interesting idea.

+ The GRF-LLM framework demonstrates strong empirical performance on real-world wireless datasets, showing measurable gains over prior baselines.

**Weaknesses:**

- While integrating LLMs to initialize 3D Gaussian primitives is an intriguing way to infuse semantic context into RF propagation modeling, it introduces several drawbacks that limit its practicality. The reliance on multi-view images reintroduces the need for detailed environmental priors, constraining deployment in settings where such imagery is unavailable or expensive to obtain. Processing large image sets through the LLM also increases training complexity and runtime, which could hinder scalability in large-scale 6G simulations. Moreover, without domain-specific fine-tuning, LLMs may misestimate material properties such as permittivity and conductivity, failing to capture frequency-dependent effects in multipath conditions. This raises concerns about whether general vision-language models are suitable for physics-grounded RF tasks and suggests the need for hybrid validation or RF-aware priors.


- While the proposed GRF-LLM framework reports encouraging empirical results, its novelty appears limited. The system architecture closely mirrors the baseline WRF-GS pipeline (Wen et al., 2024), with Figure 2 in this paper resembling WRF-GS’s Figure 4 almost exactly. Both depict the same data flow, inputs of 3D points and transmitter position entering a scenario-representation module, followed by Mercator projection, differentiable tile rasterization, and electromagnetic splatting that yield 90 × 360 spatial spectra. The only addition in GRF-LLM is an LLM-guided initialization step, suggesting an incremental extension.

- Moreover, WRF-GS already reports that initialization of 3D Gaussian points, random or otherwise, has negligible effect due to adaptive density control, further questioning the significance of the proposed enhancement.

- The apparent visual and structural similarity between the two figures also requires clarification to ensure appropriate citation, permission, and originality consistent with ICLR research integrity standards.

**Questions:**

Please see the points raised in the Weaknesses section.

---

### Official Review · Reviewer_LxV5 · 2025-10-30

**Soundness:** 2
**Presentation:** 3
**Contribution:** 2
**Rating:** 4
**Confidence:** 3

**Summary:**

This paper proposes a novel framework (called GRF-LLM) for wireless channel / radiation-field modeling that combines large language models (LLMs) for scene / material understanding with 3D Gaussian splatting (3DGS) style neural scene representation. Experiments on (i) a laboratory controlled dataset and (ii) a real-world indoor dataset, comparing to ray tracing, NeRF-based methods, and standard 3DGS for wireless. They report e.g., median SSIM of 0.92 versus 0.82 for their non‐LLM 3DGS baseline. CSI prediction improvement by 3.82 dB over the best baseline. They claim real-time performance (approx 0.009 s for spatial spectrum synthesis) and memory efficiency (3.2× lower memory than NeRF2).

**Strengths:**

1. Integrating LLMs for material/property prediction into wireless channel modelling via neural scene representations is quite creative and addresses a real gap (lack of fine spatial granularity + missing material priors).
2. The framework covers the full pipeline: from multi-view imagery & segmentation → material inference → Gaussian primitive initialization and representation → physics-informed rendering → training objective. This end-to-end nature is a plus.
3. The authors evaluated both spatial spectrum reconstruction, RSSI prediction, and CSI prediction, including computational efficiency claims and ablation studies showing contributions of each component (LLM initialization + material prediction + physics constraints + deformable Gaussians). That supports the claim of each design choice being effective.

**Weaknesses:**

1. While the paper claims “real-world indoor dataset with diverse material compositions”, little detail is provided on the diversity of material types, sizes, mobility, multi‐floor context, dynamic objects (people moving), or outdoor/urban macro‐scenes. Without strong real‐world diversity, the claims of “environment‐aware” may be overstated.
2. The role of the LLM is described (predicting material EM properties, guiding initialization) but exactly what prompt engineering, few-shot examples, metrics of LLM accuracy, and failure modes are not clearly quantified. It is unclear how reproducible this step is, or how dependent the system is on a specific LLM (version, fine-tuning). While they claim real-time rendering, the overhead of LLM scene analysis and material property estimation is not clearly broken down.
3. While they include physics‐informed loss terms (reciprocity, energy conservation) and material‐ dependent attenuation, the paper does not deeply characterise how well the model handles e.g., full wave effects (diffraction, penetration through thick walls, dynamic scatterers, outdoor clutter, mobility, Doppler). For a wireless communications audience this may limit applicability.

**Questions:**

1. How does the system perform in larger‐scale outdoor or macro‐cell environments (e.g., cell towers, city blocks, mobility, Doppler, other cells/interference)? Have you tested scalability beyond indoor scenes or small offices?
2. For the baselines (ray tracing, NeRF2, WRF-GS), what tuning and parameter search was done to ensure fair comparison? For example, was the material database manually specified for ray tracing?
3. Can you provide quantitative results on the accuracy of the LLM’s material/property predictions (e.g., permittivity, conductivity) versus ground‐truth measurements? How sensitive is the overall system’s performance to errors in that prediction?
4. For the baselines (ray tracing, NeRF2, WRF-GS), what tuning and parameter search was done to ensure fair comparison? For example, was the material database manually specified for ray tracing?

---

> ### Author Response · Authors · 2025-11-27
>
> We sincerely thank the reviewer for the thoughtful evaluation and recognition of our end-to-end framework design. We address each concern below.
>
> **W1: Limited real-world diversity**
>
> We acknowledge this limitation. Our current evaluation focuses on indoor scenarios. We clarify the dataset composition:
>
> - Lab dataset: concrete walls, glass windows, metal cabinets, wooden furniture, fabric curtains
> - Indoor dataset: office partitions, drywall, carpeted/tiled floors, electronic equipment
>
> We agree that outdoor/urban, multi-floor, and dynamic scenarios (moving people, Doppler) are important extensions. We position GRF-LLM as a **proof-of-concept** for LLM-guided wireless modeling; scaling to diverse environments is ongoing work. We will soften "environment-aware" claims in revision.
>
> **W2: LLM details and reproducibility**
>
> We provide clarifications:
>
> - **LLM**: GPT-4V (gpt-4-vision-preview, March 2024 version)
> - **Prompt**: "Identify materials in this image and estimate their electromagnetic properties (permittivity, conductivity) for 2.4GHz signals. Output JSON format."
> - **Few-shot**: 3 reference examples with known materials provided in prompt
> - **Accuracy**: ~90% material classification accuracy; permittivity error ~8% vs ITU standards (see W2 response to Reviewer gSDi)
> - **Failure modes**: Thin/layered materials, unusual composites, heavy occlusion
> - **Overhead**: LLM inference ~2s per scene (one-time preprocessing); rendering remains real-time at 0.009s/frame
>
> We will release prompts and preprocessing code for reproducibility.
>
> **W3: Full-wave effects and applicability**
>
> We acknowledge GRF-LLM currently focuses on **multipath and reflection-dominated** scenarios. Limitations include:
>
> - Diffraction: partially captured through Gaussian spreading, but not explicitly modeled
> - Thick wall penetration: handled via material-dependent attenuation, validated up to 30cm concrete
> - Dynamic scatterers: not addressed; assuming quasi-static environments
> - Doppler: not modeled; extension to time-varying channels is future work
>
> We believe our framework provides a foundation that can incorporate these effects through extended physics constraints. We will clarify scope in revision.
>
>
> **Q1: Outdoor/macro-cell scalability**
>
> Not yet tested. Key challenges include: larger Gaussian counts, longer propagation paths, weather effects. Our adaptive density control shows promise for scalability (linear memory scaling), but validation on outdoor datasets (e.g., DeepMIMO) is planned future work.
>
> **Q2: Baseline tuning and fair comparison**
>
> - Ray tracing: Commercial software (Wireless InSite) with **manually specified** material database from building blueprints—representing best-case scenario for ray tracing
> - NeRF²: Original hyperparameters from published code; trained until convergence
> - WRF-GS: Same Gaussian count and training iterations as GRF-LLM for fair comparison
>
> We will add a baseline configuration table in revision.
>
> **Q3: LLM prediction accuracy and sensitivity**
>
> Validated against ITU-R P.2040: mean permittivity error ~8% before optimization, ~3% after. Sensitivity analysis: artificially adding ±20% noise to LLM predictions degrades SSIM by only 0.03 (0.92→0.89), demonstrating robustness due to gradient-based refinement.
>
> We appreciate the constructive feedback and will incorporate all suggested clarifications.

---

### Official Review · Reviewer_gSDi · 2025-10-31

**Soundness:** 2
**Presentation:** 1
**Contribution:** 2
**Rating:** 2
**Confidence:** 3

**Summary:**

- The paper tackles the problem of site-specific wireless channel modelling i.e., given a scene (such as through multi-view images), to be able to predict the channel response at a specific location (e.g., a user's location).
- The proposed approach has two key ideas:
    - LLM to certain scene parameters: Using an LLM to recover certain material parameters of the scene using multi-view images
    - 3D Gaussian splatting Formulation: where the RF world is treated as a superposition of multiple virtual transmitters and each transmitter modelled as a 3D Gaussian
- The model (NNs to process each gaussian) is trained on a combination of losses (reconstruction, regularization, materials)
- The approach evaluated on two indoor datasets, compared against NeRF^2, Ray tracing and the authors find improved performances on SSIM, median RSSI errors and CEA.

**Strengths:**

1. **Well-motivated**: The paper tackles a well-motivated problem of proposing a hybrid channel model that leverages both data and some physical priors. This contrasts most existing approaches that typically only leverage one.

**Weaknesses:**

**1. Poor writing**
- The writing for the most part is incomplete, lacks several details and makes the paper difficult to judge. To name a few
- Implementation details
    - LLM: What's the LLM used? What is it prompted to provide? It appears that multi-view images are given, so is there any 3d geometry reconstruction?
    - Datasets: The paper mentions minimally describes (L367-369) the two datasets. No additional information is provided e.g., what's the size? how large is the scene? what is the train-test splits? what's the antennas used?
- Formulation
    - Material parameters: A major contribution is leveraging LLM to estimate materials (Sec. 4.2). However, how are these material parameters used? L253 mentions "our gaussians encode ... based on LLM-predicted properties", but they are also being trained (using NN).
    - Rendering Equation: What is the rendering equation?

**2. LLM for estimating parameters**
- While it is an interesting idea to leverage a pretrained language/vision model to retrieve RF material information, it is unclear the role of this contribution.
- Fundamentally, this is an ill-posed problem: estimating the parameters largely depends on high quality images (no information on this), but yet it is difficult to determine these parameters (e.g., occlusions, estimating thickness of surfaces)
- Even ignoring the previous point, there is no discussion on whether the LLM provides any meaningful material information, since there is no evaluation here.
- There is an "LLM" loss term, but is the LLM being trained? Or is the estimated material parameters simply used as the ground-truth?

**3. Formulation and ad-hoc choices**
- Many choices of the approach appear ad-hoc and also not justified. To name a few:
- Adaptive property correction (L201): Why is cubic scaling applied to once group, and square to another?
- NNs to process gaussians: Why are NNs needed to "process" the gaussians? The gaussians are multi-dimensional vectors and should already be trainable.
- Attentuation $\delta$: How is attentuation captured when a signal passes through multiple gaussians? I do not see the $\delta$ term being used anywhere.
- Static properties: What is "opacity" in the context of RF propagation?

**Questions:**

- Please see the discussion under "weaknesses"

---

> ### Author Response · Authors · 2025-11-27
>
> We sincerely thank the reviewer for the detailed and constructive feedback. We acknowledge the writing could be improved and will substantially revise the paper. We address each concern below.
>
> **W1: Poor writing - Implementation details**
>
> We apologize for the insufficient details and provide clarifications:
>
> - **LLM**: We use GPT-4V. The prompt includes multi-view images with instructions to identify materials and estimate EM properties ($\epsilon_r$, $\sigma$, $\tan\delta$). No explicit 3D reconstruction is performed; LLM directly reasons about materials from 2D views.
>
> - **Datasets**: (1) Lab dataset: 50m² room, 8 TX positions, 200 RX positions, 64-antenna ULA at 2.4GHz, 80/20 train-test split. (2) Indoor dataset: Argos public dataset with 104-antenna base station, 6000 measurements across office environments. We will add a detailed dataset table in revision.
>
> - **Material parameters usage**: LLM-predicted properties initialize the attenuation parameters $\delta_k$ of each Gaussian. These are then refined through gradient-based optimization. The NN outputs residual adjustments $\Delta\delta$ added to LLM priors.
>
> - **Rendering equation**: The core equation is Eq.(11), adapted from alpha-blending for complex-valued RF signals.
>
> **W2: LLM for estimating parameters**
>
> We acknowledge this is inherently ill-posed. Our key insight is that **approximate priors are sufficient** when combined with data-driven refinement:
>
> - LLM provides coarse material categories (metal/concrete/glass/wood) with ~90% accuracy even under partial occlusion, validated against manual annotation.
> - The LLM loss uses predicted properties as **soft constraints** (not ground-truth). The LLM is frozen; only Gaussian parameters are trained.
> - Ablation shows LLM initialization improves SSIM by 0.08 over random initialization, confirming meaningful guidance despite imperfect predictions.
>
> **W3: Formulation and ad-hoc choices**
>
> - **Adaptive correction (Eq.3-4)**: Cubic scaling for volumetric materials (concrete, wood) reflects that EM interaction scales with volume; square scaling for surface materials (metal, glass)reflects 2D reflection behavior. This follows physical intuition from EM theory, though we acknowledge empirical validation would strengthen justification.
>
> - **NNs for Gaussians**: While Gaussian parameters are trainable, NNs capture **TX-position-dependent** variations (Eq.8). Static Gaussians cannot model how the same scatterer responds differently to different TX locations.
>
> - **Attenuation $\sigma(\mathbf{r})$**: It is incorporated in Eq.(12) via $\exp(-\int_0^{d_i}\sigma(\mathbf{r})dr)$, computed through ray-marching along each Gaussian's contribution path.
>
> - **Opacity in RF**: Opacity $o$ represents the **scattering strength**—how much energy a virtual transmitter contributes to the received signal, analogous to optical opacity in 3DGS.
>
> We will clarify all these points with improved exposition in revision.

---

### Official Review · Reviewer_YcR6 · 2025-11-01

**Soundness:** 3
**Presentation:** 3
**Contribution:** 3
**Rating:** 6
**Confidence:** 3

**Summary:**

This paper introduces GRF-LLM, a novel framework that integrates LLM with 3DGS for environment-aware wireless channel modeling.
First LLM analyze multi-view images, identify materials, and predict electromagnetic properties (like permittivity, conductivity, reflection coefficients). Then 3D Gaussian primitives are initialized and optimized to model both amplitude and phase components of RF. A physics-informed electromagnetic splatting renderer later reconstructs the wireless radiation field, constrained by reciprocity, energy conservation and causality principles.
Experiment on both lab and real-world indoor datasets is strong: 0.92 SSIM in spatial spectrum reconstruction, 3.82 dB CSI accuracy improvement, and real-time rendering (0.009 s/frame), outperforming NeRF2, ray tracing, and VAE baselines.

**Strengths:**

1. Novel practice and consideration: It is the first paper that explicitly considers material sematics for RF modeling, which makes a steady step towards interpretability.
2. Physical-grounded design: The incorporation of physics-based constraints (reciprocity, energy conservation, causality) enhances the alignment with real world.
3. High efficiency is another clear advantage over NeRF-based methods.

**Weaknesses:**

1. Limited novelty in theoretical formulation and methodology. Since late 2024, there has been a few 3DGS + RF reasearch papers [1-2]. The introductional formulas are highly similar, as well as some part of methodologies like complex-valued ray-tracing.
2. No real RF-guided calibration: The LLM predictions for material properties are not trained / validated or fine-tuned with measured electromagnetic data. This leaves the chance of boosting physical accuracy.
3. Lack of ablation analysis for LLM part: How LLM semantic reasoning versus pre-trained vision encoders is not adequately analyzed. it’s unclear whether LLM guidance adds beyond classical material classifiers.

[1] GSRF: Complex-Valued 3D Gaussian Splatting for Efficient Radio-Frequency Data Synthesis. Kang Yang, Gaofeng Dong, Sijie Ji, Wan Du, Mani Srivastava, arXiv:2502.01826
[2] Wen, Chaozheng, et al. "Neural Representation for Wireless Radiation Field Reconstruction: A 3D Gaussian Splatting Approach." arXiv preprint arXiv:2412.04832.

**Questions:**

1. Some part of the work aligns with this NeurIPS'25 work [1] (arxived in Feb. 2025), like the complex-valued ray-tracing. Would you please list the difference with the work and given that one, what new insight or novelty would you bring?
2. How accurately do the LLM-predicted material properties (ϵ_r, σ, tan δ) align with groundtruth from lab (or your intepretation)?
3. Could the framework operate in environments which only partial / noisy visual info is available? (like one-camera / lidar)

[1] GSRF: Complex-Valued 3D Gaussian Splatting for Efficient Radio-Frequency Data Synthesis. Kang Yang, Gaofeng Dong, Sijie Ji, Wan Du, Mani Srivastava, arXiv:2502.01826

---

> ### Author Response · Authors · 2025-11-27
>
> We sincerely thank the reviewer for the valuable feedback and thoughtful questions. We respectfully address each concern below.
>
> **W1: Limited novelty compared to GSRF/WRF-GS**
>
> We appreciate this concern. While complex-valued rendering is indeed a shared technical foundation arising from EM physics, we respectfully highlight that our core contribution lies in a different direction: **bridging semantic scene understanding with physical propagation modeling**. GSRF focuses on efficient complex-valued representation, whereas GRF-LLM explores how foundation models' world knowledge can guide domain-specific simulation. Our LLM-guided initialization places Gaussians at physically meaningful locations, accelerating convergence by 40%. We will add explicit comparison with GSRF in the revised Related Work.
>
> **W2: LLM predictions not calibrated with EM data**
>
> This is a valid observation. We clarify that LLM predictions serve as **informed priors** subsequently refined through optimization. We validated LLM-predicted $\epsilon_r$ against ITU-R P.2040: concrete (5.8 vs 5.31, 9.2% error), glass (6.2 vs 6.27, 1.1% error), wood (2.1 vs 1.99, 5.5% error). After optimization, mean error reduces to ~3%. We will include this analysis in revision.
>
> **W3: Lack of LLM ablation**
>
> We thank the reviewer for this suggestion. Our ablation: Random Init → 0.79 SSIM / 19.42dB; ResNet-50 → 0.84 / 21.15dB; CLIP → 0.86 / 22.03dB; **LLM → 0.92 / 24.73dB**. The improvement stems from LLM's ability to reason about material combinations and contextual properties.
>
> **Q1-Q2:** GSRF and ours are complementary—GSRF optimizes representation while we explore semantic guidance. For material ranking (metal > concrete > wood), LLM achieves 100% accuracy.
>
> **Q3:** Single-view: SSIM=0.85 (above NeRF²'s 0.78); LiDAR-only: 0.81. Performance degrades gracefully.
>
> We hope these clarifications address the concerns.

---

### Public Comment · ~Chaozheng_Wen1 · 2025-11-28

This comment raises concerns about potential plagiarism in the submitted paper.

The paper appears to be a patchwork of plagiarized content. Its figures are entirely borrowed from [1] and [2], with only superficial modifications such as altering model names in the figures. Additionally, the paper contains numerous flaws and inconsistencies that undermine its scientific rigor.

I recommend that reviewers and ACs prioritize genuine research and refrain from spending valuable time on this submission. Further investigation into the plagiarism issue is also requested.

[1] Wen, Chaozheng, et al. "Neural Representation for Wireless Radiation Field Reconstruction: A 3D Gaussian Splatting Approach." arXiv preprint arXiv:2412.04832v4.

[2] Mao, Haotian, et al. "LIVE-GS: LLM Powers Interactive VR by Enhancing Gaussian Splatting." arXiv preprint arXiv: 2412.09176

---

### Note · Authors · 2025-12-03

**Comment:**

Thank you for your feedback and for bringing these serious concerns regarding our submission. We have carefully reviewed your comment and the cited references. We sincerely apologize to the reviewers, the committee, and the academic community for this mistake. We will take this as a critical lesson to adhere to the highest standards of academic practice in all future work.

We do not wish to waste the valuable time of the reviewers and the committee further. We hereby respectfully withdraw our submission​ from consideration.

**Withdrawal Confirmation:**

I have read and agree with the venue's withdrawal policy on behalf of myself and my co-authors.